# Building and Testing PPARγ Therapeutic ELB00824 with an Improved Therapeutic Window for Neuropathic Pain

**DOI:** 10.3390/molecules25051120

**Published:** 2020-03-03

**Authors:** Karin N. Westlund, Morgan Zhang

**Affiliations:** 1Department of Anesthesiology & Critical Care Medicine, MSC10 6000, 2211 Lomas Blvd. NE, University of New Mexico Health Sciences Center, Albuquerque, NM 87131, USA; 2USA Elixiria Biotech Inc, 200 High Point Drive, Hartsdale, NY 10530, USA; 8372651@gmail.com

**Keywords:** neuropathic pain, anxiety, depression, thiazolidinedione, pioglitazone, rosiglitazone, PEA, central sensitization, mechanical allodynia, mitochondrial bioenergetics, mice, rats

## Abstract

Effective, non-addictive therapeutics for chronic pain remain a critical need. While there are several potential therapeutics that stimulate anti-inflammatory mechanisms to restore homeostasis in the spinal dorsal horn microenvironment, the effectiveness of drugs for neuropathic pain are still inadequate. The convergence of increasing knowledge about the multi-factorial mechanisms underlying neuropathic pain and the mechanisms of drug action from preclinical studies are providing the ability to create pharmaceuticals with better clinical effectiveness. By targeting and activating the peroxisome proliferator-activated receptor gamma subunit (PPARγ), numerous preclinical studies report pleiotropic effects of thiazolidinediones (TDZ) beyond their intended use of increasing insulin, including their anti-inflammatory, renal, cardioprotective, and oncopreventative effects. Several studies find TDZs reduce pain-related behavioral symptoms, including ongoing secondary hypersensitivity driven by central sensitization. Previous studies find increased PPARγ in the spinal cord and brain regions innervated by incoming afferent nerve endings after the induction of neuropathic pain models. PPARγ agonist treatment provides an effective reduction in pain-related behaviors, including anxiety. Data further suggest that improved brain mitochondrial bioenergetics after PPARγ agonist treatment is a key mechanism for reducing hypersensitivity. This review emphasizes two points relevant for the development of better chronic pain therapies. First, employing neuropathic pain models with chronic duration is critical since they can encompass the continuum of molecular and brain circuitry alterations arising over time when pain persists, providing greater relevance to clinical pain syndromes. Assisting in that effort are preclinical models of chronic trigeminal pain syndromes. Secondly, considering the access to nerve and brain neurons and glia across the blood–brain barrier is important. While many therapies have low brain penetrance, a PPARγ agonist with better brain penetrance, ELB00824, has been developed. Purposeful design and recent comparative testing indicate that ELB00824 is extraordinarily efficient and efficacious. ELB00824 provides greatly improved attenuation of pain-related behaviors, including mechanical hypersensitivity, anxiety, and depression in our chronic trigeminal nerve injury models. Physiochemical properties allowing significant brain access and toxicity testing are discussed.

## 1. Introduction

Neuropathic pain is a chronic pain that persists over 3 months resulting from damage to nerves and/or the nervous system itself [1]. Painful diabetic neuropathy and post-herpetic neuralgia are among the most common forms of neuropathic pain. Other causes include traumatic or postsurgical nerve injuries, spinal cord injury, stroke, multiple sclerosis, cancer/chemotherapy, Herpes zoster, and HIV infection. Neuropathic pain reportedly persists permanently in half of the patients with traumatic nerve injury [2,3]. Neuropathic pain is often accompanied by anxiety, depression, and sleep disorders. The neuropathic pain that reportedly affects as many as 26 million Americans is severe, relentless, and disabling [4,5]. It was reported that 14.7% of Chinese patients suffer from chronic pain and 9% of the entire population in Hong Kong have pain with neuropathic characteristics [6]. 

## 2. Limitations of Clinically Used and Experimental Analgesics 

Current analgesics, including opioids, are far from satisfactory in providing pain relief. Safe and effective non-addictive therapeutics remain a critical need. Pain is the primary precipitating factor that has caused opioid overprescribing and its continued overuse has led to the current opioid epidemic [7]. Morphine equivalency in opioid consumption has precipitously increased over the last decades and is complicated by comorbid persisting pain and opioid-induced hyperalgesia [8,9]. It was estimated that 22.8 million patients in Europe use prescription opioids and that half a million are dependent [10]. In 2017, over 70,000 Americans died from an overdose of synthetic opioids, including oxycodone, tramadol, and especially fentanyl [2,3]. In addition, gabapentin, the 7th-most prescribed drug for pain in the United States, is now misused to enhance an opioid high, killing over 8,000 Americans last year [11]. 

The global neuropathic pain drug market is expected to reach US $8.3 billion by 2024 [12]. While opiates are heavily overprescribed, they are of little therapeutic value for treatment of this type of pain. Research for repurposing current drugs and development of new drugs for treating neuropathic pain have been actively pursued, however, in the past 10 years there have been few new non-opioid Active Pharmaceutical Ingredients approved by the FDA, making the discovery of new non-opioid medications for this disorder imperative.

Several factors contribute to the dilemma of why potential pain therapies successful in rodent models have failed in clinical trials. The lack of better understanding of pain chronification is a major contributor and is under intense study. Chronic pain has a large central sensitization component inducing altered molecular function and brain circuitry. Experimental studies investigating repurposing of current therapeutics for neuropathic pain have found some insufficiently cross the blood–brain barrier to be effective as analgesics. Many preclinical models used for testing pain therapeutics typically reverse spontaneously after 3–4 weeks, indicating tissue healing and the recovery of nerve function. These studies are not considering the continuum of molecular and brain circuitry alterations that occur leading to chronic pain states. Convergence of increasing knowledge from preclinical studies about the multi-factorial mechanisms underlying chronic neuropathic pain, novel drug targets for exploration, and mechanisms of drug action will provide the ability to create pharmaceuticals with better clinical effectiveness. 

## 3. PPARγ

For example, re-purposing PPARγ (peroxisome proliferator activated receptor-gamma subunit) agonists as analgesics and for other clinical syndromes has been considered. PPARγ is a nuclear receptor that is widely expressed in adipose, immune cells, neurons, and glia. PPARγ is activated by a variety of endogenous fatty acid derived compounds (e.g., palmitoylethanolamide (PEA) and 15-Deoxy-Delta-12,14-prostaglandin J2 (15d-PGJ2)). PEA is available in the US as a food supplement and has been available in Italy and Spain for medical purposes for over twenty years. Related synthetic thiazolidinedione (TZD) type drugs activating PPARγ (e.g., pioglitazone and rosiglitazone) are currently prescribed as oral anti-diabetic drugs to improve lipid metabolism and reduce blood pressure [13]. Other pleitropic effects of TDZ drugs have emerged, including their oncopreventive effects. For example, pioglitazone dramatically reduced both the incidence and frequency per group of mammillary tumors in a rat model [14]. PPARγ activation was shown to provide cannabidiol-induced apoptosis of human lung cancer cells [15]. PPARγ activation has been recognized as a key anti-inflammatory modulator. A review of the relevant literature finds that TZDs downregulate pro-inflammatory cytokines, such as Interleukins (IL)-4,-5,-6, as well as interfere with production of profibrotic molecules, including platelet-derived growth factor (PDGF), IL-1, and transforming growth factor beta (TGF-β) [16]. Anti-inflammatory and anti-fibrotic actions of TZDs are indicated in experimental rat models of inflammatory bowel disease, as is a role for PPARγ agonists in the treatment of inflammatory diseases, such as systemic lupus erythematous, renal disease, atherosclerosis, brain inflammation, and pancreatitis [16]. Other studies have examined the effects of TZDs on spinal cord trauma [17,18].

In human studies, testing over many years supports the contention that PEA has analgesic actions in patients suffering from various neuropathic pain states including diabetic neuropathic pain. Interestingly, endogenous and phytocannabinoids reportedly activate PPAR isoform PPARγ in addition to PPARδ and thus likely account for some of their reported anti-nociceptive effects [19,20]. The discussion below provides evidence that PPARγ is a key therapeutic target for the treatment of pain and for repurposing PPARγ agonists as effective modulators of pain. 

## 4. Building Better Animal Models for Testing PPAR Therapeutics

Pain is a complex sensation that manifests in many different ways. A major knowledge gap exists in understanding clinical pain syndromes that are chronic rather than acute in duration. It is becoming widely apparent that the chronification of pain is accompanied over time by a continuing evolution of molecular and circuitry alterations. At a critical time point, ~3–4 weeks post nerve injury, behavioral testing in most neuropathic pain models reveals reversal of hypersensitivity indicative of tissue healing and restored nerve function. In only a few of the current models does the nerve injury and nociceptive alteration continue, inducing central sensitization and the brain circuitry changes leading to the chronification of pain. Examples of persisting preclinical models include the spared nerve injury models (SNI) [21,22] and the trigeminal nerve tie (CCI-ION, dIoN) [23,24,25] or compression models (TIC, FRICT-ION) [26,27,28]. Persisting increases in levels of cytokines may explain the differences in the persistence of hypersensitivity between acute duration and chronic duration of preclinical models [29]. It is clear that the increase in IL-1β is a factor since it increases both RNA and protein expression of CSF1 in satellite glial cells in the dorsal root ganglia peripheral nerve cells of rats with SNI. The activation of glia and cytokine release that occurs both peripherally and centrally in all preclinical models after nerve injury is correlated with level of hypersensitivity, as reviewed previously [29,30,31,32]. 

## 5. PPARγ Agonist Effects on Chronic Pain Symptoms

The systemic administration of TZD drugs has been shown to reduce peripheral neuropathic pain behaviors in several preclinical models tested with single or repeated dosing [18,32,33,34,35,36,37,38,39]. In animal studies, the PPARγ agonist, pioglitazone, remains the best characterized to date with higher efficacy than rosiglitazone for reducing painful diabetic and spinal nerve ligation neuropathies [18,37,38,39]. Pioglitazone reduces the SNI-induced increases in hypersensitivity and glial fibrillary astrocytic protein (GFAP) biomarker expression [38]. Testing revealed pioglitazone has a more rapid effect than expected for reversal of hypersensitivity, within 60 min. Thus, while pioglitazone has been shown to have transcription-dependent genomic effects indicated by the decrease in the Fos protein expression typically seen in pain models, neuropathic pain reverses rapidly even in the presence of protein inhibition by intrathecal anisomycin [38]. Along with the decrease in the glial activation biomarker genomic events, these findings indicate that a relevant non-genomic, transcription-independent effect mediated by PPARγ is providing analgesia. The effectiveness of rosiglitazone treatment given in days 1–3 after partial sciatic nerve ligation in mice was much less effective than with later treatment after 21 days [36]. The effectiveness was attributed to regulation of macrophage infiltration and a reduction in inflammatory cascades at the injury site. Intense PPARγ immunoreactivity is expressed after 3 weeks in both the spinal cord and brainstem trigeminal nerve termination sites after sciatic nerve ligation or trigeminal nerve injury [19,40].

In our mouse Trigeminal Inflammatory Compression (TIC) neuropathic chronic pain model, treatment at 8 weeks with PPARγ agonist pioglitazone (100 mg/kg) attenuates the whisker pad mechanical hypersensitivity (allodynia) that persists indefinitely in this preclinical model [40,41,42]. The specificity of pioglitazone activation was demonstrated with the use of the PPARγ antagonist, GW9662, which eliminated the effect of pioglitazone [40]. Two antagonists of the PPARα subunit tested at 8 weeks had no effect on hypersensitization [40]. However, PPARα agonists are effective in reducing nociceptive behaviors in inflammatory pain models [18]. 

There is an interplay between the physical symptoms of pain and its psychological effects involving brain limbic circuitry that are revealed after weeks of persisting pain. Hypersensitivity, anxiety, depression, and even cognitive deficit are measurable in preclinical models [18,27,28,41,42,43,44,45,46]. These pain-related characteristics closely resemble symptomatology present in many patients with chronic neuropathic pain [41]. The FRICT-ION orofacial neuropathic pain mouse model while mild enough to prevent any weight gain deficits generates measureable depression by 6 weeks (Figure 1) [28].

## 6. Building a Better Therapeutic

The clinical use of PPARγ modulators pioglitazone and rosiglitazone has revealed common adverse effects, however [13]. Toxicity study for our recently developed PPARγ agonist included once-daily treatment of rats for 14 days with ELB00824 (0, 30, 120, 450 mg/kg) [27]. The rats observed daily exhibited no signs of physical, behavioral, food intake, body weight, or water consumption abnormalities. Fifty-five hematological, blood coagulation, and serum biochemistry parameters monitored showed virtually no changes. As another issue, efficiencies of the synthetic PPARγ agonists for therapeutic use in patients with Alzheimer’s may be limited somewhat by their nerve and brain permeability [47]. 

The following provides details toward design, synthesis, and testing of a brain penetrant PPARγ molecule suitable as a pain therapy more accessible to brain and nerve PPARγ binding sites [27]:
(1)Computer-Assisted Drug Virtual Screening on a high-performance computer workstation was used to search libraries of 520,000 small molecules from the Zinc15 database in order to identify 100 structures that were most likely to bind to PPARγ, with high in silico predicted permeability to the blood–nerve barrier (BNB). Then, the leads were designed to modify the chemical structure of a known compound and a product with much greater blood–brain barrier permeability was synthesized.(2)The next step was an in vitro screen of the compounds with our PPRE Luciferase reporter/PPARγ expressing Combo cell line. (3)The resultant product, ELB00824, was purified by recrystallization to 95% purity and then its structure confirmed by 1H NMR (Figure 2A). The computer-generated image shows the tight binding of ELB00824 (grey structure) with PPARγ (pink ribbon).The complete details of the ELB00824 synthesis are provided in our recent paper [27]. The ELB00824 therapeutic (USA Elixeria BioTech Inc) is protected by the international patent pending (PCT/CN2019/072302). (4)ELB00824 crosses the blood–brain barrier with high efficiency tested in conventional preclinical models of chronic neuropathic pain and reduces pain-related behaviors in both male and female rodents [27]. The ELB00824 pharmacologically targets the PPARγ proteins in immune cells and glia to reverse established peripheral nerve injury-induced nociceptive hypersensitivity. Exceptionally high in vivo blood–brain barrier (BBB) permeability among the current PPARγ agonists was confirmed by a comparative measurement of the drug key pharmacokinetic parameters in rat brain and plasma. The PPARγ activation in a stable cell line identified the half maximal effective concentration for ELB00824 (EC_50_ is 4.7 μM, Figure 2B) was higher than that of the clinically used PPARγ agonists (rosiglitazone 6.5 µM and pioglitazone 22.1 µM). 

Brain concentration (C_max)_, relative brain bioavailability denoted by Area Under the Curve (AUC_0–24_)_,_ and the brain–plasma ratio (K_p_) of ELB00824 (i.p.) were 4.43 µg/mL, 33.59 µg.h/mL, and 5.1, respectively, analyzed by HPLC [16]. This was significantly increased compared to those of pioglitazone (i.p.) by 6.5-,11-, and 58-fold, respectively, and compared to those of rosiglitazone (i.p.) by 8.7-, 17-, and 61-fold (all data, *p* < 0.01) (Figure 2C). In comparison, less than 1% of orally administered PEA is found in the brain [48]. On the contrary, the relative brain bioavailability (AUC_0–24h_) for oral vs. the i.p. injection route of ELB00824 is 31%, indicating unprecedented high brain (or nerve) bioavailability of ELB00824, i.e., one or two orders of magnitude higher than pioglitazone [27].

## 7. The Unprecedented Therapeutic Window and BBB Permeability

Drugs, treated as foreign molecules by the BBB, are unable to pass. In fact, over 98 percent of small molecule drugs do not show useful activity in the brain due to poor or no penetration of the BBB [49,50]. Therefore, the distribution of compounds between blood and brain is a very important for the consideration of new candidate drug molecules for neurological disorders. Many models for the prediction of passive blood–brain partitioning, expressed in terms of logBB or log of observed permeability surface area (logPS) values, were derived with a good predictive power [51,52]. For a given compound, a logBB > 0.3, or logPS > −1 is considered to readily cross the BBB, while a logBB < −1 or logPS < −2 are poorly distributed to the brain. A total of 49 PPARγ agonists used in clinical trials or marketed were collected and compared with ELB00824. These PPARγ agonists were ranked according to the predicted equilibrium blood–brain solute distribution (logBB) value in Table 1. See Appendix A for more details.

Table 1 shows that the logBB and logPS value of ELB00824 is highest (0.747 and −0.79, respectively), indicating that ELB00824 is the PPARγ agonist with the best BBB permeability, followed by Sodelglitazar (whose Phase 2 studies were terminated prior to completion due to safety). Note that pioglitazone, the most widely used PPARγ agonist in neurology clinical studies, has predicted poor BBB permeability (BB = 10^−0.561^, since logBB = −0.561). This is only 1/20 that of ELB00824, with its BB = 10^[0.747 − (−0.561)] = 20.

Differences of BBB permeability make a significant difference in clinical trials. Some doctors and pharmacists formerly have given high doses of drugs to patients thinking the PPARγ agonists with less BBB permeability could be used to treat neurological disease as long as they increased the dose <10-fold. However, unfortunately, numerous clinical trials have then failed. Why? Although the reasons were generally not reported, the most logical reason is that current PPARγ drugs have limited therapeutic windows in neurological disease treatment, i.e., a range of doses at which a medication is effective without unacceptable adverse events.

Using pioglitazone as an example, the minimal effective dose (MED) with efficacy in acute trigeminal neuropathic pain is 1000 mg/kg in mouse studies [27], equivalent to 4880 mg/kg in human [56]. The latter is 65-fold that of the maximum tolerated dose (MTD, i.e., 75 mg/kg [57]), thus MED >> MTD. Obviously, such a high dose of pioglitazone can not be used to treat pain without serious side effects, including upper respiratory tract infection, edema, and hypoglycemia, cardiac failure, bone fracture, headache, and pharyngitis [58]. One the contrary, the MED of ELB00824 for trigeminal neuropathic pain in mouse and rat is only 3–10 mg/kg, equivalent to 15–49 mg/kg in human. This is lower than the MTD of ELB00824 (75 mg/kg). Assuming that the MTD of pioglitazone is the same as the MTD of ELB00824, MED < MTD. It is much easier to find the therapeutic window for ELB00824. For example, the 45 mg/kg dose of ELB00824 is effective without unacceptable adverse events.

In conclusion, the excellent BBB permeability of ELB00824 provides a reasonable therapeutic window for neurological diseases, while other PPARγ agonists likely do not.

## 8. Efficacy of ELB00824 for the Reduction of Chronic Neuropathic Pain

ELB00824 reduces mechanical allodynia after persisting distal infraorbital nerve injury (dIoN) in rats of both sexes [27]. With either i.p. or p.o. dosing, the minimally effective dose of ELB00824 in the CCI pain model is 10 mg/kg, while an equivalent dose of pioglitazone was 1000 mg/kg. Topical cream application is highly efficacious (5 mg in 70 mg cream). A combination of ELB00824 (3 mg/kg) with low dose gabapentin (0.3 mg/kg) further reduces allodynia to baseline.

In comparisons to the clinical anti-anxiety therapy carbamazepine, ELB00824 was highly effective in the reduction of mechanical hypersensitivity tested in the TIC mouse chronic trigeminal nerve injury model (Figure 3) [27]. Effectiveness was demonstrated with testing on both the ipsi- and contralateral whiskerpads. The TIC model induction method here used an intraoral approach and was recently renamed “FRICT-ION” [28]. 

## 9. Effects of ELB00824 on Anxiety Measures

Anxiety-like behaviors were also assessed using the zero maze and light/dark preference tests in mice with persisting allodynia in week six post model induction. ELB00824 was more effective in the light/dark place preference test (Figure 4) [27]. In two of four anxiety measures, treatment with ELB00824 was as effective as carbamazepine in restoring the zone transition number and reducing the latency to enter the lighted area of the test chamber. In the zero maze, little effect was noted for treatment with ELB00824 with the small sample size tested (Figure 5) [27].

## 10. Role of ELB00824 in Mitochondrial Bioenergetics in Neuropathic Pain

Improvement in the mitochondrial signaling pathway and antioxidant capacity has been shown with pioglitazone treatment. An age-induced decrease in free radical reducing activity was reversed by pioglitazone (3 mg/kg, 4 weeks), which provided the enhancement of antioxidant molecule gene and protein expression levels in an atrial fibrillation model induced by 30 s burst pacing [59]. The demonstration of the improvement in the antioxidant capacity and inhibition of mitochondrial apoptotic signaling pathway was shown by dosing pioglitazone (10 mg/kg) 2 h prior to 45 min renal artery clamp and 24 h reperfusion [60].

Neuronal mitochondrial dysfunction and the associated nitro-oxidative stress are also crucial final common pathways of neuropathic pain [44,61,62]. There are many reactive oxygen species (ROS) production sites in cells, and most of them are inside mitochondria. See the red stars in the Figure 6. In the state of rest, most ROS are generated from two different sites in complex I, one site in complex II, complex III, and ETF (ETF:Q oxidoreductase). Under some circumstances, GPDH (Glycerol 3-phosphate dehydrogenase) or OGDH (Oxoglutarate Dehydrogenase) will also produce ROS [39]. Through catalyzation by an enzyme called nitric-oxide synthase or NOS, ROS are able to form peroxynitrite (ONOO^−^), which is a type of reactive nitrogen species (RNS). Since our preliminary study found ELB00824 and pioglitazone inhibit ROS and RNS production (unpublished data), it was proposed that they may interact directly with mitochondrial targets.

Some mitochondrial targets have been identified (Figure 6), such as complex I of the respiratory chain [63], MitoNEET (an iron transport protein), or MPC (mitochondrial pyruvate carrier) [64]. The respiratory control rate (RCR) is the best general measure of mitochondrial function. A high RCR implies that the mitochondria have a high capacity for substrate oxidation and ATP turnover, a low proton leak, and low ROS generation. We found that the RCR was compromised in the cortex of mice with the persisting TIC nerve injury [44]. Combined low doses of pioglitazone (100 mg/kg) and D-cycloserine (80 mg/kg), an antibiotic used to treat tuberculosis, given daily for 7 days improved mitochondrial bioenergetics. 

Other analgesics only provide symptomatic treatments, since they only target certain downstream effectors in transduction (e.g., ion channels) after their upregulation, or the latter steps in the pain process after the release of ROS [61]. Antioxidants interact directly with ROS to remove it. This removal may be too late in the processes that lead to chronic pain. Compared with antioxidants, PPARγ agonists, e.g., pioglitazone, do not directly scavenge ROS [13]. This indicates that ELB00824 may also target upstream effectors (e.g., by dissembling complex I) to prevent ROS generation, rather than direct ROS removal. However, the mitochondrial targets of ELB00824 are not yet identified and will be part of our on-going projects. 

The preclinical testing of the non-opioid, ELB00824, has revealed the following advantages: (1)**Extraordinarily efficient and efficacious**. ELB00824 has the highest CNS penetration among existing PPARγ agonists. Our animal studies indicate that the analgesic effect of our ELB00824 is more highly efficient than other PPARγ agonists (100 times than Pioglitazone, over 30 times than PEA), and some clinically used non-opioid analgesics (over 10 times than Carbamazepine). Effectiveness was rapid and sustained longer than pioglitazone.(2)**Highly safe**. ELB00824 is one of few non-opioid analgesics with excellent skin permeability, and thus topical ELB00824 provides excellent anti-allodynic effects with far better benefit/risk ratios than other non-topical agents. In addition, our animal study demonstrated that it is completely safe, even at the dose of 45 times of the effective dose. Furthermore, it may be non-addictive, since there is no record of addiction in the long history of the usage of any PPARγ agonist drug or supplements (e.g., PEA).(3)**Provides both etiological and symptomatic treatment.** ELB00824 can provide comprehensive neuroprotection through its target protein PPARγ, a master gatekeeper in nerve injury and repair. These neuroprotection actions include improvement in the aggravated oxidative stress and inflammation condition, both of which are root causes of neuropathic pain. On the contrary, opioids and other non-opioid analgesics do not protect nerves.

Testing of potential PPARγ therapeutics in suitable models persisting for longer durations to better mimic clinical syndromes is essential to development of analgesics that will provide effective pain relief. An added advantage of these models is the ability to test the relief of pain-related anxiety and depression. Further development of the non-opioid ELB00824 analgesic with brain and nerve penetrability capable of effectively restoring neuronal function and improving mitochondrial bioenergetics is a critical need.

## Figures and Tables

**Figure 1 molecules-25-01120-f001:**
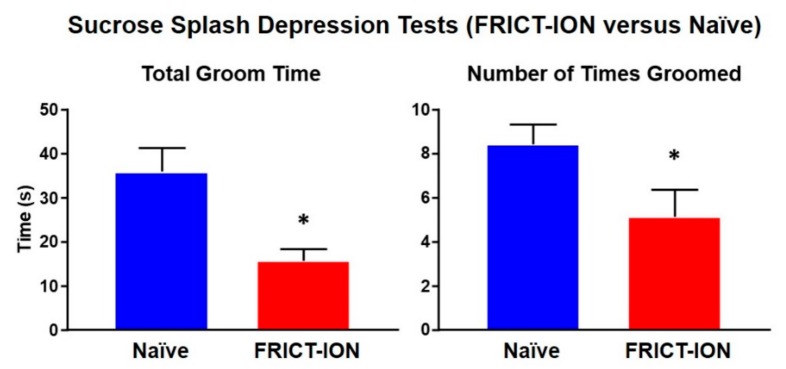
Depression assessed by spraying 10% sucrose on the rump of mice results in grooming behaviors in naïve male mice but significantly less grooming in mice with neuropathic pain induced by FRICT-ION model at 6 weeks. The asterisk indicates *p* < 0.05 compared to naïve (*n* = 4).

**Figure 2 molecules-25-01120-f002:**
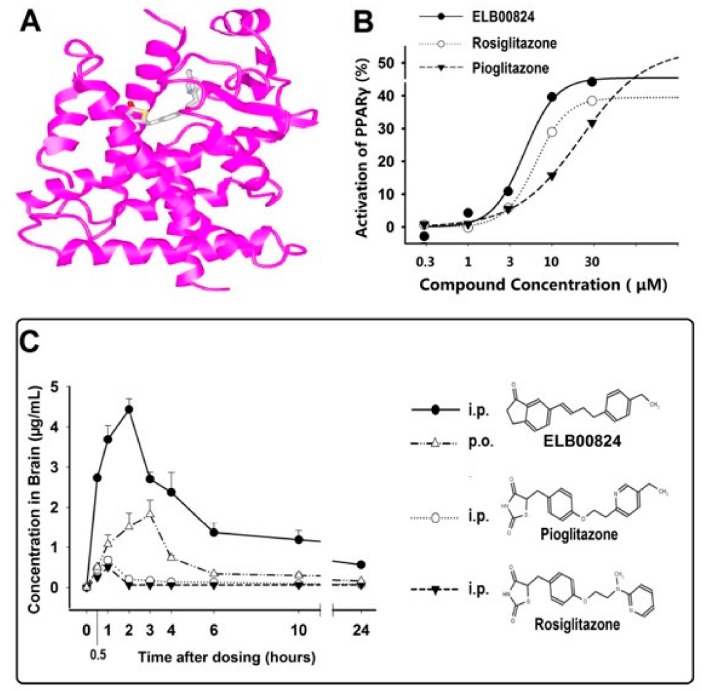
(**A**) Nuclear magnetic resonance (NMR) structural confirmation of EL00824. ELB00824 (grey structure) binds tightly with PPARγ (pink ribbon). (**B**) ELB00824 exhibits the highest in vitro PPARγ transcriptional activity compared to clinically used PPARγ agonists. (**C**) EL00824 is absorbed rapidly and the peak plasma and brain concentrations (Cmax) reached at 1 and 3 h, respectively. The chemical structures of EL00824, pioglitazone, and rosiglitazone are shown at the side for comparison [27].

**Figure 3 molecules-25-01120-f003:**
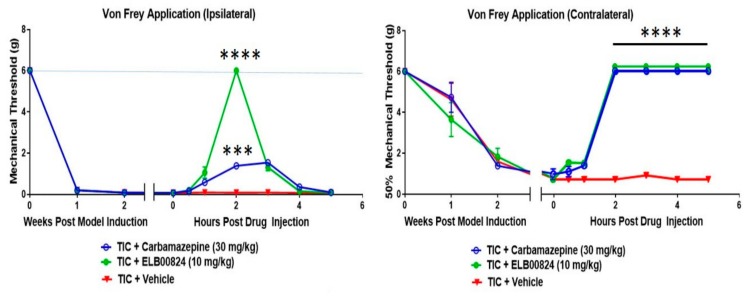
Efficacy of ELB00824 and carbamazepine was tested after injection (i.p.) on day 21 post Trigeminal Inflammatory Compression (TIC) nerve injury in male mice. The mechanical threshold was determined 0, 0.5, 1, 2, 3, 4, and 5 hours after drug injection (*n* = 4) testing on both ipsi- and contralateral whiskerpads. The data are presented as a mean ± SEM. Two-way ANOVA with Dunnett’s multiple comparisons test. For all graphs, asterisks (***) indicate p < 0.001 and (****) *p* < 0.0001 compared to vehicle. [27].

**Figure 4 molecules-25-01120-f004:**
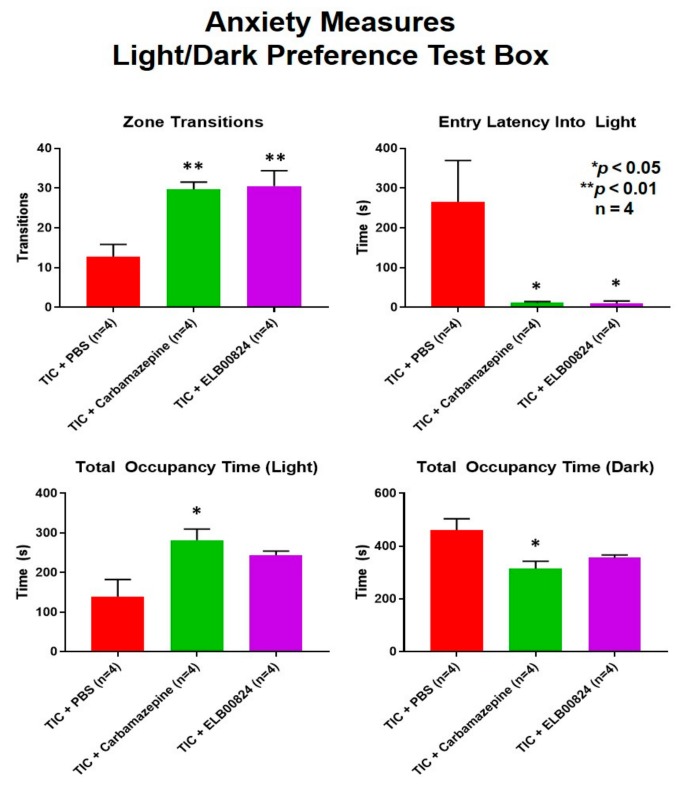
Anxiety develops in rodent neuropathic pain models persisting longer than 6 weeks. ELB00824 (10 mg/kg) significantly attenuated anxiety in the mice with chronic TIC at six weeks, in two of the light/dark place preference test measures. Measures included zone transitions, entry latency into the lighted chamber, the total occupancy time in the lighted chamber, and the total time in the dark chamber. The decrease in anxiety was equivalent to a reduction by carbamazepine (30 mg/kg) for at least two of the measures. The asterisks (*) indicate *p*  <  0.05 and double asterisks (**) indicate *p*  <  0.01 compared to TIC with vehicle (*n*  =  4) [27].

**Figure 5 molecules-25-01120-f005:**
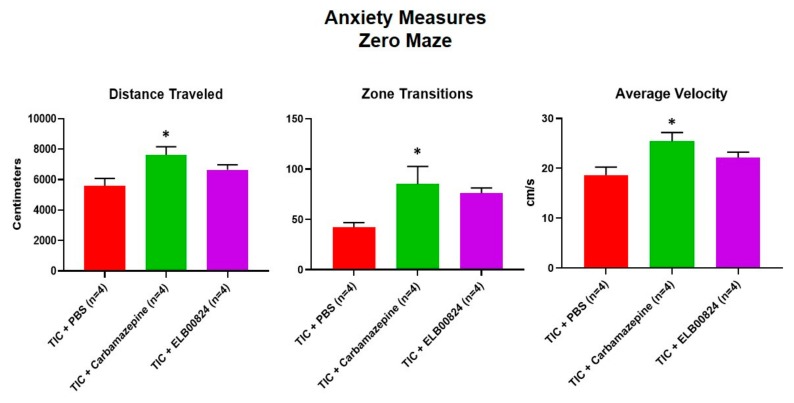
Efficacy of ELB0824 was less than that of carbamazepine with zero maze anxiety testing. The testing of ELB00824 was done in mice with TIC chronic neuropathic pain persisting 6 weeks. The asterisks (*) indicate *p* < 0.05 compared to TIC with vehicle (*n* = 4) [27].

**Figure 6 molecules-25-01120-f006:**
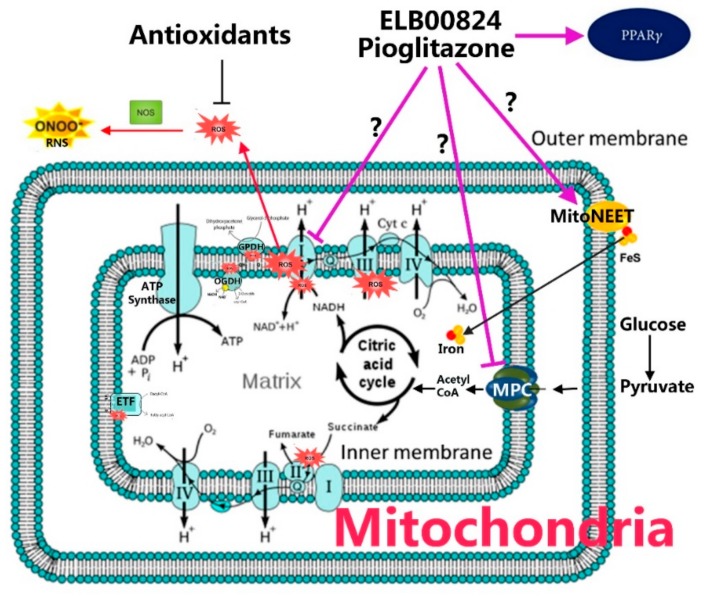
Major mitochondrial sites of reactive oxygen species (ROS) production as potential targets of ELB00824 and antioxidants. The red stars are major ROS production sites. It is hypothesized that ELB00824 targets mitochondrial proteins, such as complex I of the mitochondrial respiratory chain, mitoNEET, or MPC (mitochondrial pyruvate carrier), and leads to a reduction in ROS generation, while antioxidants target ROS directly.

**Table 1 molecules-25-01120-t001:** Currently available PPARγ agonists and their Blood–Brain Permeability (BBB).

Developer	Compound Name	MW	LogD_7.4_*	pKa of Acid Group*	TP SA*	N + O	BBB Permeability		
							Log BB**	Log PS***	Conclusion
Elixiria	ELB00824	290	5.94	N/A	17.1	1	0.747	−0.79	Excellent
GlaxoSmithKline	Sodelglitazar	500	3.67	3.58	59.4	4	0.408	−1.76	Good
Emerald Health	VCE-​004.8	434	6.20	N/A	66.4	4	0.208	−1.16	Fair
CymaBay	Arhalofenate	416	3.88	N/A	64.6	5	0.178	−1.75	
Many companies	Astaxanthin	597	8.05	N/A	74.6	4	0.036	−0.75	
Mitsubishi Tanabe	Netoglitazone	381	3.82	6.40^A^	55.4	4	0.030	−1.69	
Zydus Cadila	Saroglitazar	440	2.23	3.73	60.7	5	−0.030	−2.15	Poor perme- ability
PPM Services	GED-0507-34-levo	195	−2.04	3.90	72.6	4	−0.074	−3.38	
Badische	Tetradecylthioacetic acid	288	3.32	4.58	37.3	2	−0.148	−1.66	
BioPhytis	Norbixin. Macuneos	380	0.42	4.71, 5.31	74.6	4	−0.177	−2.75	
Many companies	Daidzein	254	1.77	6.48	66.8	4	−0.182	−2.33	
Chipscreen	Chiglitazar	573	5.03	3.57	80.6	6	−0.223	−1.60	
Merck Sante	Oxeglitazar	314	0.72	4.30	55.8	4	−0.287	−2.50	
Glaxosmithkline	Farglitazar	547	4.03	3.76	101.7	7	−0.289	−2.05	
GlaxoSmithKline	GSK-376501	532	2.97	3.35	79.2	7	−0.343	−2.12	
Dainippon Sumitom	DSP-8658	417	2.68	3.96	68.5	5	−0.429	−2.10	
Kyorin	MK-0767	422	2.22	6.40^A^	93.7	7	−0.430	−2.45	
Lilly	Etalocib	545	5.18	3.60	85.2	6	−0.516	−1.60	
Aestus	FK-614, ATx08-001	468	3.84	3.94	81.1	6	−0.542	−1.91	
Omeros, Takada	OMS-403, Pioglitazone	342	2.17	6.40^A^	68.3	5	−0.561	−2.24	
Complexa	10-nitrooleic acid	327	3.59	4.99	80.4	5	−0.564	−1.97	
Daiichi Sankyo	Troglitazone	442	5.29	6.40, 10.8^A^	84.9	6	−0.567	−1.57	
Daiichi Sankyo	Rivoglitazone	397	2.12	6.40^B^	82.5	7	−0.573	−2.38	
Bristol-Myers Squibb	Peliglitazar	531	1.77	3.26	111.3	9	−0.582	−2.73	
Takeda	Imiglitazar	471	2.20	4.15	94.2	7	−0.604	−2.46	
Daiichi Sankyo	Efatutazone	503	3.97	6.40^A^	108.5	8	−0.672	−2.13	
Many companies	Mesalazine	153	−1.72	2.02	83.6	4	−0.689	−3.39	
Minoryx	MN-102	358	0.87	6.40^A^	88.5	6	−0.714	−2.76	
T3D	T3D-959	421	1.77	4.20	81.8	6	−0.718	−2.46	
Novo Nordisk	Ragaglitazar	419	1.73	3.73	68.2	6	−0.718	−2.35	
Pfizer	Darglitazone	420	3.00	6.40 ^A^	89.3	6	−0.722	−2.21	
Glaxosmithkline	Rosiglitazone	357	2.83	6.10, 6.80 ^C^	71.5	6	−0.727	−2.09	
Japan Tbacc	Reglitazar	392	3.40	6.40 ^A^	90.7	7	−0.751	−2.12	
Bristol-Myers Squibb	Muraglitazar	517	1.34	3.18	111.3	9	−0.778	−2.84	
Lilly	Naveglitazar	422	1.55	3.64	74.2	6	−0.833	−2.45	
Roche	Edaglitazone	465	4.05	6.40 ^A^	81.4	6	−0.872	−1.86	
Lilly	LY-510929	464	2.45	3.63	81.8	6	−0.877	−2.28	
Dr Reddy’s	Balaglitazone	395	1.67	6.40 ^A^	88.1	7	−0.906	−2.55	
Roche	Aleglitazar	438	1.79	4.30	81.8	6	−0.958	−2.46	
Pfizer	Indeglitazar	389	−0.60	3.53	94.8	7	−1.018	−3.20	Very poor perme- ability
Eisai	E-3030	436	1.65	3.99	88.8	6	−1.089	−2.56	
Daiichi Sankyo	DS-6930	389	1.26	3.85	97.3	7	−1.128	−2.74	
AstraZeneca	Tesaglitazar	408	−0.15	3.73	99.1	7	−1.136	−3.12	
Coherus, InteKrin	CHS 131	514	6.15	7.11	68.3	5	−1.167	−1.19	
Takeda	Sipoglitazar	464	2.10	3.87	86.5	7	−1.230	−2.42	
Abbvie	Lanifbranor	435	0.75	3.76	89.3	6	−1.327	−2.80	
Theracos	CLX-0921	520	4.52	6.40 ^A^	100.2	8	−1.331	−1.91	
Merck	MK-0533	527	3.29	3.82	87.0	7	−1.362	−2.11	
Chong Kun Dang	Lobeglitazone	490	3.48	6.40 ^A^	102.9	9	−1.466	−2.20	
Novartis	Cevoglitazar	559	1.62	2.86	109.9	8	−1.592	−2.75	

*Data were predicted by MarvinSketch software (version 6.1.4, Chemaxon, Boston, MA, USA), unless specified otherwise. **calculated by software pkcsm [51]. ***calculated by the model: logPS = −2.19 + 0.262 × logD_7.4_ + 0.0583 × vsa_base − 0.009 × TPSA [52], vsa_base is 0 for all molecules and TPSA is the topological polar surface area. (N + O): the sum of the nitrogen and oxygen. A: the pKa is estimated from the thiazolidinedione group of pioglitazone, at [53]. B: the pKa is from [54]. C: the pKa is from [55].

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
