# Peer review of "Building and Testing PPARγ Therapeutic ELB00824 with an Improved Therapeutic Window for Neuropathic Pain"

_molecules, 2020, doi:10.3390/molecules25051120_

Round 1

Reviewer 1 Report

Manuscript molecules-704149 by Westlund K.N., seems to be focused on recent developments of PPARgamma agonists evaluation for the therapy of chronic pain. However, going into the review it appears that this work substantially deals with the identification and preclinical evaluation of ELB00824, with extensive referring to the authors' experimental work (ref.19) which is actually under review.

Therefore in my opinion this is not a review on the subject declared by the title, this is a manuscript sponsoring ELB00824 potentiality and usefulness, and repeating what it has been just submitted ( reference 19)

Author Response

Thank you for the thorough reviews. We have re-written and re-arranged our review. Many additional references have been added along with new review text shown in blue font.

Manuscript molecules-704149 by Westlund K.N., seems to be focused on recent developments of PPARgamma agonists evaluation for the therapy of chronic pain. However, going into the review it appears that this work substantially deals with the identification and preclinical evaluation of ELB00824, with extensive referring to the authors' experimental work (ref.19) which is actually under review. Therefore in my opinion this is not a review on the subject declared by the title, this is a manuscript sponsoring ELB00824 potentiality and usefulness, and repeating what it has been just submitted ( reference 19)

Response: The reference [19] to the new animal model under review is one of the tools we used to assess ELB00824 but the main aim of this review is to demonstrate the effectiveness and advantages of this new PPARγ agonist therapy over current FDA approved PPARγ drugs when used as an analgesic. The ELB00824 is better by far than other available PPARγ agonists tested by us and others in consideration of these drugs for new use as neuropathic pain analgesics. This review provides additional information about the physiochemical/pharmacological properties of ELB00824.

Reviewer 2 Report

The manuscript entitled “Building and Testing Better PPARγ Therapeutics for Neuropathic Pain“ provides interesting and clinically relevant topic potentially very interesting for readers. The manuscript is presented at the desirable scientific level suitable for this prestigious journal.

Minor point: line 86, please describe more precisely the pleiotropic effect of TZD in organism. I strongly suggest to mention well-described anticancer 1, anti-inflammatory 2, antioxidant 3, anti-angiogenic 4, and anti-thrombotic effects 5 of TZD.

Suggested references:

Bojkova et al. Eur J Cancer Prev. 2010 Sep;19(5):379-84 (dramatic decrease in cancer risk in vivo!) Vetuschi et al. Eur Rev Med Pharmacol Sci. 2018 Dec;22(24):8839-8848. Xu et al. J Cardiovasc Electrophysiol. 2012 Feb;23(2):209-17. Rudnicki et al. Eur J Pharmacol. 2016 Jul 5;782:98-106. Liu et al. Mol Pharmacol. 2016 Feb;89(2):313-21.

Remark for authors: Please shortly explain to Editors, what is the novelty of this paper compared to paper referenced No.19 (under review) or vice versa, because Figs 2-5 are taken from mentioned unpublished article.

Author Response

Thank you for the thorough reviews. We have re-written and re-arranged our review. Many additional references have been added along with new review text shown in blue font.

Comments and Suggestions for Authors

The manuscript entitled “Building and Testing Better PPARγ Therapeutics for Neuropathic Pain“ provides interesting and clinically relevant topic potentially very interesting for readers. The manuscript is presented at the desirable scientific level suitable for this prestigious journal.

Minor point: line 86, please describe more precisely the pleiotropic effect of TZD in organism. I strongly suggest to mention well-described anticancer 1, anti-inflammatory 2, antioxidant 3, anti-angiogenic 4, and anti-thrombotic effects 5 of TZD.

Response: Done

Suggested references:

Bojkova et al. Eur J Cancer Prev. 2010 Sep;19(5):379-84 (dramatic decrease in cancer risk in vivo!) Vetuschi et al. Eur Rev Med Pharmacol Sci. 2018 Dec;22(24):8839-8848. Xu et al. J Cardiovasc Electrophysiol. 2012 Feb;23(2):209-17. Rudnicki et al. Eur J Pharmacol. 2016 Jul 5;782:98-106. Liu et al. Mol Pharmacol. 2016 Feb;89(2):313-21.

Response: Thank you for these vital references that are in fields outside our purview. We have added most of these and many others to the review.

 Remark for authors: Please shortly explain to Editors, what is the novelty of this paper compared to paper referenced No.19 (under review) or vice versa, because Figs 2-5 are taken from mentioned unpublished article.

Response: Actually, the two figures we have re-used are from our published Ref #27. This use in reviews is allowed through policy of the journal Molecular Pain since we hold/purchased the copyright. The manuscript was published in August with full details of many behavioral studies showing reduction of experimental neuropathic pain and anxiety in both rats and mice. The citation has been added to the figure legend. This review provides more information about the ELB00824 itself and the behavioral data is necessary to support the utility of ELB008244 in another clinical syndrome. Other references in our field of study are also cited. Our model detailed in ref #19, is currently under review but has few references here in this review.

Reviewer 3 Report

Whilst the manuscript is well written and results look good I don't see how it can be considered a "Review". Please resubmit as a research paper focused on ELB00824. 

Furthermore, as the co-author Morgan Zhang is affiliated with the company Elixiria, who are the rights holders over the drug ELB00824, the conflict of interest statement does not make any sense???? How could there not be a conflict of interest. 

Author Response

Comments and Suggestions for Authors

Thank you for the thorough reviews. We have re-written and re-arranged our review. Many additional references have been added along with new review text shown in blue font.

Please resubmit as a research paper focused on ELB00824. 

Response: The re-organization and the additional text/references should now qualify this as a "Review". The additional references have been added in response to Reviewer #2.  Table 1, Figures 1, 6 and part of Figure 2 are original to this manuscript. Please see Ref#18.

 Furthermore, as the co-author Morgan Zhang is affiliated with the company Elixiria, who are the rights holders over the drug ELB00824, the conflict of interest statement does not make any sense???? How could there not be a conflict of interest. 

Response: Affiliations and conflict statements have been updated.

Reviewer 4 Report

The authors describe the importance of PPARgamma as a therapeuthic target for chronic pain. The issue is relevant and timely need. However, the manuscript raises a number of serious concerns:

Lack of qualified references for statments such as: 'In 2017, 17,029 americans died...' (line 40); ...killing 8,000 Americans last year.' (line 44); 'In the US... for this disorder imperative' (lines 56-61). Line 66: there are a number of animal models that cause long-lasting hypersensitivity (mechanical and/or thermal) for more than 3-4 weeks. Authors must find them. There are extensive literature on that matter. Lines 68-69: there is no reference, at all, suporting the statment '...therapeuthics including analgescs are developed to exclude access across blood-brain-barrier and cannot affect the neurons and glia altered'. This is is just not true.   Lines 111-112: It is not true that psychological effects under chronic pain are revealed only after 6-8 weeks of pain. Again, there extensive literarure showing that depression and/or anxiety-like behaviors are observer much earlier. Lines 136-163: the authors described a number of results (Fig.2 A,B, and C) from a manuscript that are under review elsewhere (reference 19). Therefore, I recommend this manuscript rejection.

Author Response

Thank you for the thorough reviews. We have re-written and re-arranged our review. Many additional references have been added along with new review text shown in blue font.

Comments and Suggestions for Authors

The authors describe the importance of PPARgamma as a therapeuthic target for chronic pain. The issue is relevant and timely need. However, the manuscript raises a number of serious concerns:

Lack of qualified references for statments such as: 'In 2017, 17,029 americans died...' (line 40); ...killing 8,000 Americans last year.' (line 44); 'In the US... for this disorder imperative' (lines 56-61). Line 66: there are a number of animal models that cause long-lasting hypersensitivity (mechanical and/or thermal) for more than 3-4 weeks. Authors must find them. There are extensive literature on that matter.

Response: additional references have been added, particularly in the first Introductory paragraph referred to.

 Lines 68-69: there is no reference, at all, suporting the statment '...therapeuthics including analgescs are developed to exclude access across blood-brain-barrier and cannot affect the neurons and glia altered'. This is is just not true.  

Response: This sentence has been re-written to suggest that the blood-brain barrier (BBB) is the biggest challenge to delivering PIO to the brain as a treatment for Alzheimer’s disease and a reference is included.

Lines 111-112: It is not true that psychological effects under chronic pain are revealed only after 6-8 weeks of pain. Again, there extensive literarure showing that depression and/or anxiety-like behaviors are observer much earlier.

Response: Clarification is added

Lines 136-163: the authors described a number of results (Fig.2 A,B, and C) from a manuscript that are under review elsewhere (reference 19).

Correction: The results are published in Ref 27 and is allowed under the publisher’s policy for the Molecular Pain journal. We as authors hold the copyright.

Round 2

Reviewer 1 Report

Authors have quite ameliorated their work, but I have two minor revisions two suggest: Title. Currently authors have revised the original submission title and now it is" Building and Testing a PPARγ Therapeutic for Neuropathic Pain with a Better Therapeutic Window" In my opinion it should be "Building and Testing a PPARγ Therapeutic for Neuropathic Pain with Improved Therapeutic Window" Table 1. Molecular structures should be drawn with chemdraw or similar. Currently molecular structures are reported as figures and they appear deformed. Figure 2. Molecular structures are deformed.

Author Response

Reviewer#1 Authors have quite ameliorated their work, but I have two minor revisions two suggest: Title. Currently authors have revised the original submission title and now it is" Building and Testing a PPARγ Therapeutic for Neuropathic Pain with a Better Therapeutic Window" In my opinion it should be "Building and Testing a PPARγ Therapeutic for Neuropathic Pain with Improved Therapeutic Window" Table 1. Molecular structures should be drawn with chemdraw or similar. Currently molecular structures are reported as figures and they appear deformed. Figure 2. Molecular structures are deformed.

Response: Title is revised and the molecular structures have been revised by the editorial staff I believe. They appear much better in the last version.

The title is now shown as “Building and Testing PPARγ Therapeutic ELB00824 with Improved Therapeutic Window for Neuropathic Pain”

Reviewer 3 Report

Whilst it does appear that a significant conflict of interest exists with the authors being inventors and patent holders of the molecule that is chiefly described in the review that conflict is now disclosed. 

Although some original data does appear in the review the authors have correctly pointed out that it is mostly from a previous publication of there lab. 

The authors rightly point out some deficits in preclinical pain testing with models not properly modelling the chronicity of pain in the clinic which is a serious and important issue in the field of pain research. 

The development and testing of the new BBB/BNB permeable PPARy agonist ELB00824 is impressive and the review is well written and broadly covers the field of research. It is still a concern that some bias may be given to the drug since the authors are inventors but the comparative data is well referenced and appears fair. 

In general the review is easy to read and informative. 

Minor points:

Line 237 - brackets should be (4) not (3)

Line 243 - there is an open ended bracket

Line 281 - some example references could be inserted of the trials in question. 

Author Response

Thank you.

Reviewer#3 Whilst it does appear that a significant conflict of interest exists with the authors being inventors and patent holders of the molecule that is chiefly described in the review that conflict is now disclosed. 

Response: Yes

Although some original data does appear in the review the authors have correctly pointed out that it is mostly from a previous publication of there lab. 

Response: Yes, we hold the copyright. Table 1 is a new compilation from the literature and websites for this review.

The authors rightly point out some deficits in preclinical pain testing with models not properly modelling the chronicity of pain in the clinic which is a serious and important issue in the field of pain research.

Response: Yes 

The development and testing of the new BBB/BNB permeable PPARy agonist ELB00824 is impressive and the review is well written and broadly covers the field of research. It is still a concern that some bias may be given to the drug since the authors are inventors but the comparative data is well referenced and appears fair. 

Response: Yes. Table 1 (and new Suppl. Material and Table S1 are new compilation from the literature and websites for this review.

In general the review is easy to read and informative. Thank you.

Minor points:

Line 237 - brackets should be (4) not (3)  

Response: Amended

Line 243 - there is an open ended bracket

Response: Amended and another one found was also amended.

Line 281 - some example references could be inserted of the trials in question. 

Response: Amended and expanded.

Reviewer 4 Report

The review doesl not contribute significantly to the field.

Author Response

Fortunately, the other reviewers and the editor do not concur with this opinion.